

# Basal Water Storage Variations beneath Antarctic Ice Sheet

Jingyu Kang[1,2], Yang Lu[1], Yan Li[1,2], Zizhan Zhang[1] and Hongling Shi[1]

[1]State Key Laboratory of Geodesy and Earth's Dynamics, Innovation Academy for Precision Measurement Science and Technology, Wuhan 430071, China.

[2]University of Chinese Academy of Sciences, Beijing 100049, China

*Correspondence to*: Jingyu Kang (kangjingyu17@mails.ucas.ac.cn)

**Abstract.** Antarctic basal water storage variations (BWSV) contain basal water migrations and basal melting. Identifying these variations are critical to understand the behaviour of ice sheet, yet it is rarely accessible to direct observation. We presented a layered gravity density forward/inversion method for constructing Antarctic basal mass balance (BMB) estimates

from multisource satellite observation data, and evaluated BWSV based on basal melting rate. As an example, spatial annual BWSV trend during 2003-2009 are estimated. Results reveal spatial variability of BWSV, with the rate of 46.3 Gt/y. Similar spatial distribution between basal water increases regions and locations of active subglacial lakes indicates that basal water storage in most active subglacial lakes are increasing. Comparison of spatial BWSV and ice surface velocity display a positive correlation between considerable basal water decreases and rapid ice flows, however, exceptions are when the

massive rapid ice flows connected to huge ice shelves that hold up by surrounding terrains, that slows down the basal water discharge outward.

## 1 Introduction

Antarctic basal water is widely generated by geothermal heating, basal pressure melting and frictional heating. The basal water converge into subglacial lakes or spread over ice-bed interface, and the scattered waters are connected to each other

(Wingham et al., 2006;Fricker et al., 2016), which forms basal drainage systems with complex basal water migrations (Pattyn, 2008;Carter et al., 2015). Fluctuation in basal water storage influence basal effective pressure and trigger changing ice sheet velocities (Bell and Robin, 2008;Fricker et al., 2007;Alley, 1992).

Temporal variations in basal water storage are controlled by basal melting rate and basal water migrations. Basal ice melt in high basal melting rate regions to replenish basal water storage, while basal water refreezes when flowing through

supercooling regions to reduce the storage, besides, basal water migrations between basal drainage systems and oceans also influence spatial basal water storage variation. To date, many studies have been conducted on Antarctic basal condition (Alley et al., 1998;Rignot and Jacobs, 2002;Augustin et al., 2007;Fisher et al., 2015;Martos et al., 2017;Liefferinge et al., 2018), particularly, Pattyn (2010) inferred the Antarctic basal melting rate using a hybrid ice sheet/ice stream model.





Consequently, a key question in exploring basal water storage variation (BWSV) is to evaluate the basal mass balance (BMB) 30 mainly caused by basal water migrations.

Although studies have been performed on the surface ice sheet and ice shelves, BMB over the ice sheet remains poorly understood due to limited observation. The commonly adopted approach is to examine the local surface height variations provided by altimetry observations, by assuming surface height variations related to subglacial lakes discharge (Flament et al., 2014;Wingham et al., 2006). However, this method may be invalid if BMB occur in regions without sufficient surface 35 expressions or with complex basal conditions, such as the Siple Coast (Young et al., 2016). Göeller et al (2012) proposed a balanced water layer concept in large scale ice sheet models to present BMB, while the modelled results depend greatly on the reliability of data used to build models.

In this paper, we adopted an elaborate gravity forward modelling and surface density discrimination method in combination with satellite gravity data, satellite altimetry data, GPS data and relevant models to separate the initial gravity values caused 40 by BMB, then layered gravity density inversion method was employed to estimate the final values of BMB trend. Afterward, the initial values of BMB were used to address the gravity variation caused by surface mass change and ice sheet's vertical movement, by assuming that BMB can be expressed on surface height variation. Finally, we combined BMB with basal melting rate data to evaluate BWSV. The results were also verified and interpreted based on existing research.

## 2 Methods

### 2.1 Temporal variations in Antarctic gravity field

Temporal variations in gravity field are caused by the mass redistributions within the Earth and on or above its surface. In Antarctica, these mass redistributions occur mainly on different layers such as surface of ice sheet, ice-bed interface and interior of the Earth. Accordingly, we can express the Antarctic integrated time-variable gravity field as a superposition of layered gravity variations:

$$dg_{Ant} = dg_{surf} + dg_{BMB} + dg_{solid} + dg_{ivm} \qquad (1)$$

where $dg_{Ant}$ is Antarctic integrated time-variable gravity field that can be observed by gravity satellite mission; $dg_{surf}$ is gravity variations caused by Antarctic surface process, such as snow accumulation/sublimation, surface runoff and ice flow (Ligtenberg, 2014); $dg_{BMB}$ is gravity variations caused by basal mass balance (BMB), such as basal water migrations between basal drainage systems and oceans; $dg_{solid}$ describes the mass exchange of solid Earth, such as mantle convection 55 (Fig. 1); $dg_{ivm}$ is additional gravity variation caused by the BMB-induced ice sheet's vertical movement. All the gravity variations above are defined in geoid.

In order to estimate BMB for Antarctic ice sheet, $dg_{BMB}$ should be extracted from the integrated time-variable gravity field (Equation 2), then BMB can be calculated based on gravity inversion method.





$$dg_{BMB} = dg_{Ant} - dg_{surf} - dg_{solid} - dg_{ivm} \tag{2}$$

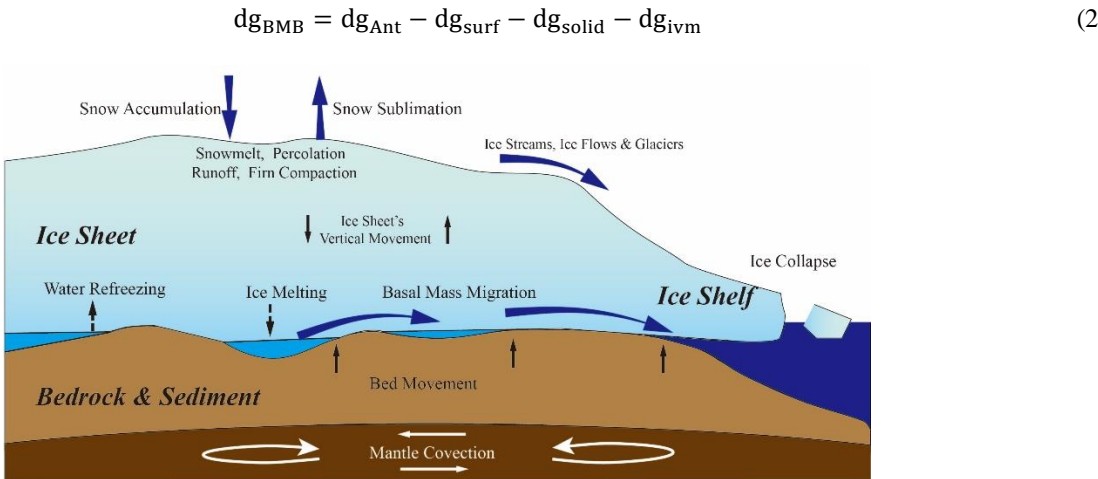

**Figure 1** Mass redistribution and movement of Antarctica

In Equations 1&2, $dg_{solid}$ is obtained from glacial isostatic adjustment (GIA) model that describes the Earth's isostatic deformation with the load on the crust since the last glacial stage(Tushingham and Peltier, 1991;Peltier, 2002). $dg_{surf}$ and $dg_{ivm}$ are calculated by utilizing gravity forward modelling method and corresponding density data, providing the known height variations caused by surface process and ice sheet's vertical movement. However, height variations caused by surface mass changes is complicate, especially in ice flow regions where the surface height variations dominated by ice thinning, and the combination of surface and ice density is more appropriate in gravity forward modelling process; furthermore, BMB will cause ice sheet's vertical movement, and this movement is also expressed as height variations and result in non-mass change induced gravity variation $dg_{ivm}$. These coupled variations are contained in gravity and surface height variations that captured by satellite gravity and altimetry mission.

**2.2 Surface density discrimination and ice sheet's vertical movement correction**

When utilizing gravity forward modelling method to calculate Antarctic gravity variations in Equation 2, the key is to separate the height variations in different layers and the ice sheet's vertical movement from altimetry observation data. We presented a surface density discrimination and ice sheet's vertical movement correction method, combined with satellite altimetry data and firn densification model (FDM) (Ligtenberg et al., 2011;Gunter et al., 2014), to estimate the corresponding height and gravity variation. The Antarctic surface height variations $dh_{Ant}$ observed by satellite altimetry can be expressed as follows:

$$dh_{Ant} = dh_{acc} + dh_{me} + dh_{fc} + dh_{if} + dh_{ivm} + dh_{cvd} \tag{3}$$

where $dh_{acc}$, $dh_{me}$ and $dh_{fc}$ are height variations caused by surface snow accumulation, melting and firn compaction respectively. These variations can be derived from FDM that provides temporal surface height variations caused by the surface mass balance variations, liquid water processes (snowmelt, percolation, refreezing and runoff) and firn compaction



(Ligtenberg et al., 2011;Gunter et al., 2014). $dh_{if}$ are height variations caused by ice flow. $dh_{ivm}$ are the ice sheet's vertical movement caused by BMB. $dh_{cvd}$ are bed movement caused by crustal viscoelastic deformation (Fig. 1).

For calculating $dg_{surf}$, surface density discrimination method is adopted in gravity forward modelling process, and the corresponding surface density $\rho$ is assigned as follows:

$$\rho = \begin{cases} \rho_{firn} \text{ for } dh_{FDM}, \ \rho_{ice} \text{ for } (dh_{surf} - dh_{FDM}) & \text{if } |dh_{surf} - dh_{FDM}| > 2\sigma_{dh} \\ \rho_{surf} \text{ for } dh_{surf} & \text{otherwise} \end{cases} \quad (4)$$

where $\rho_{firn}$ is firn density (Ligtenberg et al., 2011), $dh_{FDM} = dh_{acc} + dh_{me} + dh_{fc}$ refers to the surface height variations derived from FDM, $dh_{surf} = dh_{Ant} - dh_{cvd} - dh_{ivm}$ refers to height variations only caused by surface process, $\rho_{ice}$ is ice density (917 kg m$^{-3}$) and $\sigma_{dh} = \sqrt{\delta_{Ant}^2 + \delta_{FDM}^2}$.

As shown in Equation 4, the height variations caused by surface process are assumed to contain two processes: firn densification and ice flow. When calculating $dg_{surf}$, height difference between $dh_{surf}$ and $dh_{FDM}$ greater than $2\sigma_{dh}$ are attributed to ice flow (that is, $dh_{if} = dh_{surf} - dh_{FDM}$), and ice density $\rho_{ice}$ is assigned; the remaining $dh_{FDM}$ are attributed to firn densification process, and firn density $\rho_{surf}$ is assigned. Otherwise, the surface height variations are assumed to arise from only firn densification process (that is, $dh_{if} = 0$).

Then, the initial values of $dg_{BMB}$ can be obtained from Equation 2, and equivalent water height (EWH) of BMB can also be calculated through gravity inversion method. Afterward, we used the EWH to replace ice sheet's vertical movement $dh_{ivm}$ in Equation 3 and recalculated the surface gravity variations $dg_{surf}$ using surface density discrimination method. For the corresponding gravity correction of ice sheet's vertical movement $dg_{ivm}$, we used gravity forward modelling method, combined with EWH of BMB to calculate the gravity variations caused by the whole Antarctic ice sheet before and after it moves. The ice sheet surface and basal height data used in gravity forward/inversion method comes from BEDMAP2 (Fretwell et al., 2013) and LITHO1.0 (Pasyanos et al., 2014). Accordingly, the updated value of $dg_{BMB}$ is calculated using Equation 2, and corresponding EWH of BMB can also be calculated through gravity inversion method.

Afterward, we combined BMB and basal melting rate data to evaluate basal water storage variations (BWSV). At the base of Antarctic ice sheet, regions with the basal melting rate greater than 0 are assumed to be able to maintain basal water migrations, and the total BWSV are the gross of BMB and basal ice melt. While regions with the melting rate of 0 are assumed to be at supercooling condition, and BMB in these regions are assumed to be caused by non-aqueous mass variation. Then basal water storage variations $dM_{BWSV}$ and non-aqueous mass variations $dM_{non}$ are expressed as follows:

$$\begin{cases} dM_{BWSV} = dM_{BMB} + dM_{melt} & dM_{melt} > 0 \\ dM_{non} = dM_{BMB} & dM_{melt} = 0 \end{cases} \quad (5)$$

where $dM_{BMB}$ is basal mass balance calculated by the above gravity forward/inversion method, $dM_{melt}$ is basal melting rate for Antarctic ice sheet that comes from Pattyn (2010).





**2.3 Gravity density inversion method based on forward modelling**

In Equations 1&2, Antarctic integrated time-variable gravity data $dg_{Ant}$ comes from gravity satellite mission, other gravity variations come from related observation and modelling data that provide variation of potential coefficients, or volume variations and related density data. In order to convert the volume variations and density data into gravity variations, we adopted an oblique triangular prism gravity forward modelling method that address terrain surfaces on ellipsoids without gaps. In this method, the surface of the Earth was divided into many equilateral triangles with sides approximately 50km, and the gravity variations for each oblique triangular prism are given by the basic gravity forward modelling equation (Hofmann-Wellenhof and Moritz, 2006):

$$dg = -G\rho_v \iiint \frac{z}{(x^2+y^2+z^2)^{3/2}} dxdydz \tag{6}$$

where G is the gravitational constant, $\rho_v$ is the density related to the volume variations.

In order to calculate the equivalent water height (EWH) of BMB, we developed a layered gravity density inversion method. In this method, basal mass variation was assumed to occur on a thin layer overlying Antarctic bedrock, with a fixed thickness and variable density. Given a gravity point in Antarctic geoid, the gravity variation caused by BMB could be calculated by the integration of the modified triangular prism described as follows:

$$dg_{BMB} = \sum_{k=1}^{N} G\rho_k [\iint \frac{1}{(x_k^2+y_k^2+z_k^2)^{\frac{1}{2}}} dx_k dy_k - \iint \frac{1}{(x_k^2+y_k^2+(z_k+dH)^2)^{\frac{1}{2}}} dx_k dy_k] \tag{7}$$

$$\rho_k = \rho_w H_k \tag{8}$$

where N is the number of triangular prisms within the integrated radius, dH is the thickness of the fixed layer on Antarctic bedrock, $\rho_w$ is water density, $\rho_k$ is the density of the fixed thin layer, and $H_k$ is the EWH height of BMB.

Then, Equation 7 could be modified as follows:

$$\frac{dg_{BMB}}{G\rho_w} = \sum_{k=1}^{N} H_k [\iint \frac{1}{(x_k^2+y_k^2+z_k^2)^{\frac{1}{2}}} dx_k dy_k - \iint \frac{1}{(x_k^2+y_k^2+(z_k+dH)^2)^{\frac{1}{2}}} dx_k dy_k] \tag{9}$$

For M points in the Antarctic basal layer, Equation 9 can be expressed as:

$$d = BX \tag{10}$$

where

$$d = \begin{bmatrix} \frac{dg_{BMB_1}}{G\rho_w} \\ \vdots \\ \frac{dg_{BMB_M}}{G\rho_w} \end{bmatrix}; B = \begin{bmatrix} T_{11} & \cdots & T_{1N} \\ \vdots & \ddots & \vdots \\ T_{M1} & \cdots & T_{MN} \end{bmatrix}; X = \begin{bmatrix} H_1 \\ \vdots \\ H_N \end{bmatrix};$$






$$T_{MN} = \left[ \iint \frac{1}{\left(x_{mn}^2 + y_{mn}^2 + z_{mn}^2\right)^{\frac{1}{2}}} dx_{mn} dy_{mn} - \iint \frac{1}{\left(x_{mn}^2 + y_{mn}^2 + (z_{mn}+dH)^2\right)^{\frac{1}{2}}} dx_{mn} dy_{mn} \right]$$

Then the conjugate gradient method was employed to solve the EWH of BMB (that is, X in Equation 10).

## 3 Data processing

### 3.1 Time-varying gravity data

The Antarctic integrated time-variable gravity data $dg_{Ant}$ in Equations 1&2 comes from the dual-satellite Gravity Recovery

and Climate Experiment (GRACE) mission, which provides accurate estimates of the Earth's time-variable gravity field with a spatial resolution of approximately 300 km (Wahr et al., 1998;Wahr et al., 2004;Zhang et al., 2009). In this study, the GRACE data spanning from February 2003 to December 2009 (excluding June 2003) comes from the Department of Earth Observation and Space Systems (DEOS) Mass Transport model 1b (DMT-1b), a joint effort of the Delft University of Technology (The Netherlands) and the Global Navigation Satellite System Research and Engineering Center of Wuhan

University (China). This model was calculated on the basis of K-band ranging data from the GRACE satellite mission, and the strip error was eliminated owning to the implementation of a statistically optimal Wiener-type filter based on full covariance matrices of the noise and signal in the post processing (Liu et al., 2010).

The monthly time-variable gravity field data employed in this paper is consistent with the Antarctic altimetry mission period in order to avoid sampling effects. Ellipsoidal correction was performed on spherical harmonic coefficients to address the

Earth ellipsoidal effect (Moritz, 1980;Rapp, 1981;Heck and Seitz, 2003), and W12a (Whitehouse et al., 2012) model was used to account for gravity variations caused by solid Earth (that is, GIA, $dg_{solid}$ in Equations 1&2 ). Then Antarctic integrated time-variable gravity $dg_{Ant}$ was calculated as follows:

$$dg_{Ant} = \frac{GM}{R^2} \sum_{n=2}^{max}(n-1)\frac{1}{1+k_n} \sum_{m=0}^{n} \overline{P}_{nm}(\cos(\theta))[\Delta C\_e_{nm}\cos(m\phi) + \Delta S\_e_{nm}\sin(m\phi)] \qquad (11)$$

where GM is the geocentric gravitational constant, R is the mean radius of Earth, $k_n$ is the load Love number of degree $n$

(Wahr et al., 1998), $\overline{P}_{nm}$ are the normalized associated Legendre functions, $\Delta C\_e_{nm}$ and $\Delta S\_e_{nm}$ are modified ellipsoidal harmonic coefficients, $\theta$ and $\phi$ are colatitude and longitude, and $r(\theta)$ is the geocentric distance related to $\theta$. The annual trend of integrated gravity variations during 2003-2009 are shown in Figure 2a.

### 3.2 Antarctic altimetry and crustal viscoelastic deformation

The Antarctic altimetry data comes from the Ice, Cloud, and Land Elevation Satellite (ICESat) mission, which was launched

on January 13, 2003, at Vandenberg Air Force Base in California and was managed by the NASA Earth Observation System (EOS). The ICESat GLAS (Geoscience Laser Altimeter System) data from 18 of the campaigns were employed in this study,





spanning from February 2003 to December 2009. Dr. Hongling Shi processed ICESat data through crossover analysis method. The inter-campaigns biases were eliminated by the height comparison in regions of interior East Antarctica where the snowfall close to zeros (Shi, 2010). Figure 2b displays the annual Antarctic height variation trend $dh_{Ant}$ derived from

ICESat, with the spatial grid of $100\times100$ km$^2$. Green colours refer to apparent mass loss that are mainly located in regions around the Pine Island/Thwaites Glaciers along Amundsen coast, Jutulstraumen Glacier, Totten Glacier, and Mercer ice flow; Red colours represent apparent mass increase that are mainly distributed in Palmer Land of Antarctic Peninsula, Kamb Ice Stream in West Antarctica, and interior East Antarctica.

In our study, crustal viscoelastic deformation ( $dh_{cvd}$ in Equation 3) comes from W12a predicted basal uplift rate

(Whitehouse et al., 2012). For the consistency in experimental period, we forced the predicted basal uplift rates to the GPS observed uplift rates during 2003-2009 (Sasgen et al., 2016) (Fig. 2c).

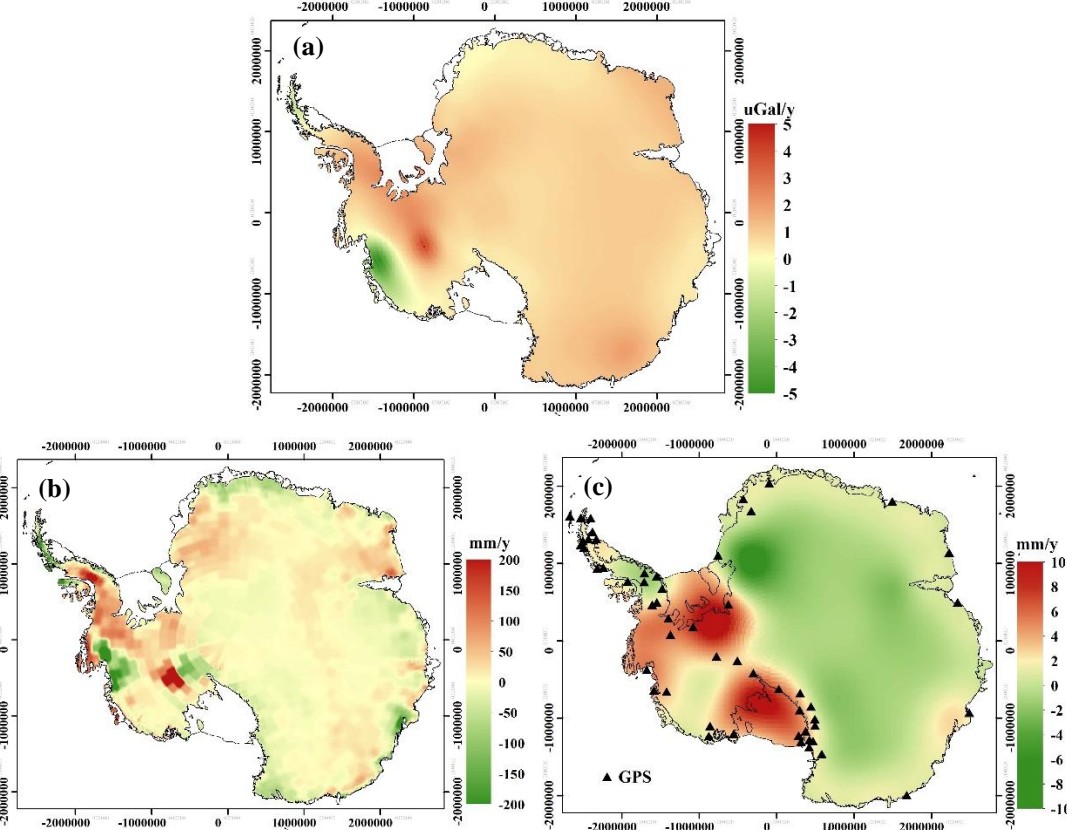

**Figure 2(a)** Gravity variations derived from DMT-1b, spanning a period from February 2003 to December 2009 (truncated at 5 uGal/y), the sampling is consistent with ICESat mission period. **(b)** Surface height variations observed by ICESat from 2003 to 2009 (truncated at 200 mm/y), this data was processed by Dr. Hongling Shi (Shi, 2010). **(c)** Antarctic bedrock uplift rate predicted by W12a (truncated at 10 mm/y), black





triangles are locations of GPS sites employed in this study.

# 4 Results and discussion

## 4.1 Antarctic basal water storage variations

According to the layered gravity forward/inversion method, we calculated Antarctic BMB and combined with basal melting
rate data to estimate the annual trend of Antarctic basal water storage variations (BWSV). Error estimation of BWSV is
complex due to the combination of multi-source data in gravity forward/inversion process. We used cylinder model
(Heiskanen and Moritz, 1967) combined with error propagation law to estimate the error of BWSV. Major error sources in
estimating BWSV are listed as follows:

$$m_{BWSV} = \sqrt{m_{GRACE}^2 + m_{GIA}^2 + m_{surf}^2 + m_{BM}^2} \tag{12}$$

Where $m_{GRACE}$ are standard deviations of GRACE. These data are provided in forms of internally calibrated standard
deviations of spherical harmonic coefficients from DMT (Liu et al., 2010). We estimated the total gravity variations errors
through method of Wahr et al. (2006); $m_{GIA}$ are GIA model errors in forms of error intervals of spherical harmonic
coefficients that are provided by Whitehouse et al. (2012); $m_{surf}$ are uncertainties related to Antarctic surface process that
mainly contains surface height variation (ICESat) and FDM; $m_{BM}$ refers to standard deviations of basal melting rate (Pattyn,
2010;Van Liefferinge and Pattyn, 2013).

Figure 3a provides equivalent water height (EWH) of BWSV, where red colours represent basal water increases regions
where the input volume of basal water is greater than the output volume; the green colours represent regions with basal water
decreases. Figure 3b provides the standard deviations of BWSV.





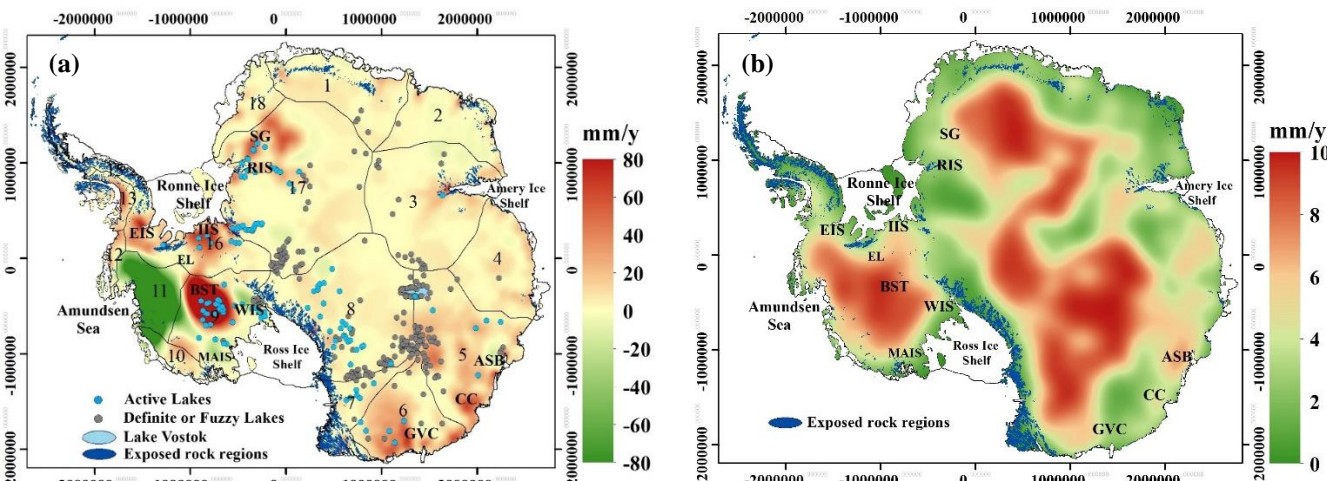

**Figure 3(a)** basal water storage variations (truncated at 80 mm/y). **(b)** standard deviations.

ASB = Aurora Subglacial Basin; BST = Bentley Subglacial Trench;

CC = Clarie Coast; EL = Ellsworth Lake; EIS = Evans Ice Stream;

IIS = Institute Ice Stream; GVC = George V Coast; MAIS = MacAyeal Ice Stream;

RIS = Recovery Ice Stream; SG = Slessor Glacier; WIS = Whillans Ice Stream;

As shown in Figure 3a, Antarctic basal water increases occur mainly in the Bentley Subglacial Trench (BST), George V

Coast (GVC), Clarie Coast (CC), Recovery Ice Stream (RIS), Slessor Glacier (SG), Institute Ice Stream (IIS) and Evans Ice

Stream (EIS) regions, indicating that there are suitable basal conditions for water storage in these regions. Among them, BST

region possess the most obvious basal water increases, this is attributed to the low basal hydraulic head that facilitates the

collection of surrounding basal water. As a result, the increased basal water in BST region will cause the uplift of the

overlying ice sheet, and this uplift has been verified herein through the analysis of CryoSat-2 data in this region (Fig 30,

(Göeller, 2014)). Similar to the case in BST region, basal water increases in the GVC and CC regions should be also caused

by low basal hydraulic head that conduces to the collection of surrounding waters. In RIS region, the basal water increases

arise from the water in recovery lakes discharging into the bedrock trench underneath the Recovery Glacier (Fig. 4, (Göeller,

2014), Fig. 1, (Bell et al., 2007)). While the basal water increases in EIS and IIS regions are caused by the blocking effect of

the Ronne Ice Shelf and their surrounding terrains that slows down the basal water discharge outward.

Basal water decreases shown in Figure 3a occur mainly along Amundsen Sea coast region, in upstream of Whillans Ice

Stream (WIS) and MacAyeal Ice Stream (MAIS) regions, and in Aurora Subglacial Basin (ASB) region. Specifically, the

significant basal water decreases along Amundsen Sea coast region are caused by the actively melting in response to basal

geothermal flux, which lead to a majority of basal water discharging into Amundsen Sea through basal channels (Fig. 1,

(Schroeder et al., 2014)). The basal water decreases in this region show similar magnitude and spatial distribution with the

mass loss previously evaluated through satellite gravity and altimetry method (Chen et al., 2009;Groh et al., 2014;Gunter et

al., 2009). This indicate that the considerable mass loss in this region is dominated by basal water discharging rather than





surface mass loss, and the surface height decreases observed by satellite altimetry are mainly due to the basal-water-loss-induced ice sheet sinking. The basal water decreases in upstream WIS and MAIS regions are caused by the basal water flow outward through basal flow paths in basal hydrological system (Fricker et al., 2016;Gray et al., 2005). In Ellsworth Lake (EL)

region, the decreases is caused by the basal water discharging into Amundsen Sea (Göeller, 2014). In ABS region, the basal water decreases are consistent with Wright's result (Fig. 6, (Wright et al., 2012)): in his study, the regional airborne geophysical survey was combined with numerical ice sheet modelling to reveals that the ASB possess extensively distributed basal pressure melting, and the basal hydraulic gradient drives basal water discharge into ocean through the basal flow paths, thereby resulting in basal water decreases in this region.

Figure 3a further shows similar spatial distribution between basal water increases and active subglacial lakes (blue circles) that observed by surface height changes (Smith et al., 2009), such as the active subglacial lakes in BST, IIS, RIS, SG and GVC regions. This indicates that the basal water volume in most of active subglacial lakes are increasing, despite water drainage occurs frequently. While the definite or fuzzy lakes (grey circles) (Carter, 2007;Andrew and Martin, 2012) are distributed mainly in interior East Antarctica with low basal water variations.

Moreover, we found that considerable basal water decreases in our result are often accompanied by rapid ice flow, such as Pine Island Glacier, Thwaites Glacier, Whillans and MacAyeal Ice Stream in WA and Totten Glacier in East Antarctica (Fig. 26a, (Göeller, 2014)). This correlation is consistent with the viewpoint of Bell and Robin (2008): the active basal water movement contribute to lubricate ice-rock interface and speed up the ice flow above. The accelerated ice flow also contributes to the basal water migrations below it, causing more basal mass decreases in these regions. However, basal water

decreases become slight or even increase, when the downstream ice flows are connected to immense ice shelves that hold up by surrounding terrains (such as the ice flow around the Ronne Ice Shelf and Amery Ice Shelf). This is attributed to the blocking effect of the ice shelf and its surrounding terrain that slows down the basal water discharge outward.

We also assessed the BWSV values and standard deviations in 18 drainage basins (Rignot et al., 2019). This drainage basin division method is performed based on the spatial similarity of simulated basal meltwater pathways and surface ice flow.

The results are listed in Table 1.





**Table 1: Antarctic basal water storage variations rates and standard deviations of 18 drainage basins**

| Basin | Rate (Gt/y) | Std (Gt) | Basin | Rate (Gt/y) | Std (Gt) | Basin | Rate (Gt/y) | Std (Gt) |
|-------|------|------|-------|------|------|-------|------|------|
| B1 | 3.2 | 3.6 | B8 | 5.4 | 11.8 | B15 | 0 | 0.1 |
| B2 | 1 | 2.1 | B9 | 23.7 | 5.7 | B16 | 10 | 3.4 |
| B3 | 4.4 | 7.8 | B10 | -3.1 | 0.8 | B17 | 13.7 | 14.1 |
| B4 | 3.8 | 3.8 | B11 | -48.7 | 3.6 | B18 | 1 | 1 |
| B5 | 15.1 | 8.7 | B12 | 0.7 | 0.4 | | | |
| B6 | 12.1 | 4.4 | B13 | 1.7 | 0.5 | | | |
| B7 | 1.9 | 2.0 | B14 | 0.4 | 0.1 | Total | 46.3 | 24.3 |

As shown in Table 1, there are five drainage basins (B5, B6, B9, B16 and B17) exhibit obvious basal water increases, and one drainage basin (B11) displays significant basal mass decreases; while the variations in the other drainage basins are less than 10 Gt/y. The mean BWSV rates beneath the Antarctic ice sheet is 4 mm year-1. This lead to a total volume of 46.3 Gt/y, accounting for about 71% of the basal melting rate (65Gt/y) (Pattyn, 2010), and also indicates that most of the basal melted water are influenced by basal hydrological movement.

**4.2 Antarctic basal non-aqueous mass variations**

According to Antarctic BMB and supercooling regions (where basal melting rate is 0), we estimated the annual trend of Antarctic non-aqueous mass variations. The error estimation is also performed based on equation 12. Figure 4a provides the EWH of basal non-aqueous mass variations and the location of definite or fuzzy subglacial lakes. Where red colours represent basal non-aqueous mass increase regions, in these regions, mass increases are caused mainly by basal water refreezing; the green colours represent regions with basal non-aqueous mass decreases that caused by ice discharge or glacial erosion. Figure 4b is the standard deviations. We also assessed the values of basal non-aqueous mass variations, and result shows balanced state with the total variations rate of -10 Gt/y, indicating that basal mass balance is dominated by BWSV rather than non-aqueous mass variations.



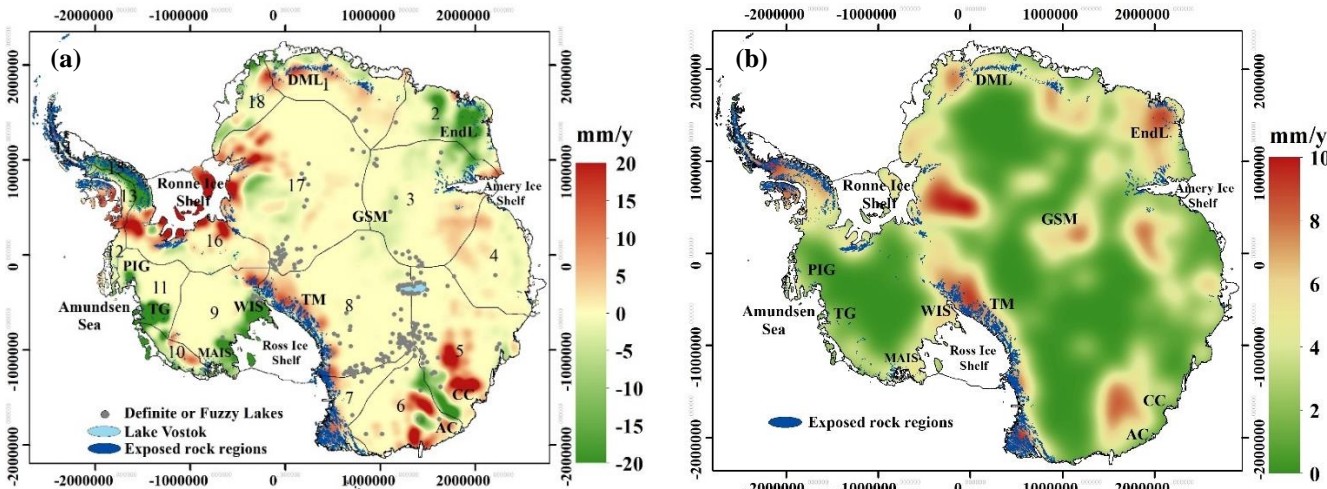

**Figure 4 (a)** basal non-aqueous mass variation (truncated at 20 mm/y). **(b)** standard deviations.

AC = Adelie Coast; DML = Dronning Maud Land; EndL = Enderby Land; GSM = Gamburtsev Subglacial Mountain

PIG = Pine Island Glacier; TG = Thwaites Glacier; TM = Transantarctic Mountains;

As shown in Figure 4a, the obvious basal non-aqueous mass variations are distributed mainly in the margin of Antarctica, with the magnitude smaller than that of basal water storage variations. Most of definite/fuzzy subglacial lakes are located mainly in regions with low basal non-aqueous mass variations.

Antarctic basal non-aqueous mass increases are situated mainly in Transantarctic Mountains (TM), Dronning Maud Land (DML), CC, east to Amery Ice Shelf regions and regions surrounding Ronne Ice Shelf, which is attributed to the basal supercooling condition that conduce to the basal water refreeze when flowing through these regions. In TM and DML regions, the widespread increases might be caused by basal hydraulic gradient that drives basal water flow upward and refreezes on basal ridges (Creyts and Clarke, 2010), or be caused by that the surface process in exposed rock regions were incorrectly estimated by FDM (espically in TM region). The former possibility of basal water flowing upward to basal ridges has been verified in the south of Gamburtsev Subglacial Mountain (GSM) through analysing a comprehensive geophysical data set (Creyts et al., 2014).

Basal non-aqueous mass decreases are located mainly in Antarctic Peninsula (AP), Pine Island Glacier (PIG), downstream Thwaites Glacier (TG) and MAIS, Enderby Land (EndL) and south of Adelie Coast (AC) regions. The decreases in WIS and MAIS regions should be attributed to the glacial erosion. While in AP and AC regions without rapid ice flow, the decreases might arise from the greater uncertainties in firn densification model (FDM) that are distributed mainly in AP, along Amundsen coast and Wilkes Land's costal region (Ligtenberg et al., 2011), and these uncertainties may lead to the underestimation in surface mass loss in gravity forward modelling process, thereby resulting in estimation errors in these regions. The decreases in TG and PIG fast surface ice flow regions might also be caused by glacial erosion or the overestimation of surface firn densification process that impede the discrimination of ice flow in forward modelling process.





Finally, basal non-aqueous mass decreases are also shown in EndL region with slow ice flow, this might be caused by the observation error in satellite altimetry data that underestimate surface height variation, or by basal water discharge outward, and verifying these possible conjectures need further effort.

**5 Conclusions**

In this study, we presented a layered gravity density forward/inversion method, allowing for ice sheet's verticals movement corrections, for constructing Antarctic basal mass balance (BMB) estimates from multisource satellite observation data. As a result, we utilized satellite gravity data, satellite altimetry data and relevant models to calculate Antarctic basal mass balance (BMB) during 2003-2009. Afterward, BMB and basal melting rate data were combined to estimate Antarctic basal water storage variations (BWSV) and basal non-aqueous mass variation separately.

Our results indicate that West Antarctica possesses considerable basal water storage variations, with the significant basal water decreases located mainly along Amundsen Sea coast, in upstream Whillans Ice Stream and MacAyeal Ice Stream regions. Along Amundsen Sea coast region, the most significant basal water decreases are caused by the active basal melting, which lead to a majority of basal water discharging into Amundsen Sea through basal channels. Accordingly, we deduced that the considerable mass loss along Amundsen Sea coast is dominated by basal water discharging rather than surface mass loss. In most regions of East Antarctica, basal water storage is increasing, with extreme values located mainly in margin regions. We found a similar spatial distribution between basal water increases and active subglacial lakes, this indicates that water in most active subglacial lakes are increasing, despite basal water drainage occur frequently. Considerable basal water decreases are often accompanied by rapid ice flow, while exceptions are when the downstream ice flows are connected to immense ice shelves that hold up by surrounding terrains, which slows down the basal water discharge outward.

On the other hand, basal non-aqueous mass decreases often occur in rapid ice flow regions, such as Pine Island Glacier, Thwaites Glacier, MacAyeal Ice Stream. These decreases are considered as glacial erosion, or the overestimation of surface firn densification process in gravity forward modelling. Extensive basal ice increases are founded in basal ridge regions with low-temperature, such as the Dronning Maud Land, and Transantarctic Mountains. We inferred that these increases might be caused by that the surface process in exposed rock regions were incorrectly estimated by FDM, or be caused by the basal hydraulic gradient in these regions can drive basal water flow backward to basal ridges. The reason for the basal non-aqueous mass decreases in Enderby Land regions remain unclear, and explaining these variations need further efforts.

**Code/Data availability**

Code and data used in this study is available at https://github.com/Kangjingyu17/Code-and-data-for-Basal-Water-Storage-Variations-beneath-Antarctic-Ice-sheet.git. ICESat data is available at https://nsidc.org/data/icesat/ (last access: 12 March 2021). BEDMAP2 data is available at https://secure.antarctica.ac.uk/data/bedmap2/ (last access:). LITHO1.0 data is available at https://igppweb.ucsd.edu/~gabi/litho1.0.html (last access: 12 March 2021). GPS data can be accessed at https://dep1doc.gfz-potsdam.de/documents/102 (last access: 12 March 2021). W12a data can be accessed at

http://www.pippawhitehouse.com/ (last access: 12 March 2021). Any other specific data and code in this study are available on request to the authors.

**Author contributions**

Kang and Lu conceived the research. Kang performed forward modelling and inversion calculation. Shi processed ICESat data. Kang, Li and Zhang collected the verification data. Kang, Lu and Li interpreted the Antarctic basal mass variation,

Kang, Lu and Zhang wrote the manuscript. All authors read and commented on the manuscript.

**Competing interests**

The authors declare that they have no conflict of interests.

**Acknowledgements**

The authors would like to thank Dr Ligtenberg, Dr Pattyn and Dr Van Liefferinge, for the use of FDM/firn density data and

basal melting rate data.

**Financial support**

This research was jointly funded by the Natural Science Foundation of China (Grant No. 41674085 and 41874093).



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
