# Peer review of "Basal Water Storage Variations beneath Antarctic Ice Sheet"

_The Cryosphere, 2021_

## Referee Comment (RC2)

This article seeks to quantify the magnitude of the trend in basal water storage across Antarctica from the analysis of satellite gravity and altimetry data. This is a novel study, which draws on a wide range of information to isolate the basal water storage component, but in the current version of the manuscript it is difficult to understand some of the methodology, there is insufficient information to enable the results to be reproduced, and without a more robust assessment of the potential limitations of the approach it is difficult to assess whether the results accurately reflect the rate of change of basal water storage across Antarctica between 2003-2009. I outline below a number of points that I feel need to be addressed to improve the clarity and robustness of the study.

**Structure and writing style**: the overall structure of the article is clear, with information presented in a logical order, but it would be useful to include more signposting, e.g. "as described in section X". The writing style is concise but there are a number of grammatical errors, which occasionally leads to some ambiguity with regard to the information being conveyed. The summary at the end of the introduction section is quite confusing on first reading. In general, the methods could be explained more clearly – see comment below. Several statements in the combined 'results and discussion' section require additional evidence, e.g. inclusion of a reference, to support the argument being made. In general, arguments are rather brief and explanations for the results lack justification.

**Methodology**: the method employed is quite complex, involving estimation of the gravity signal, surface elevation signal, or mass change signal associated with a range of processes. Satellite data and model output are used to constrain various components of the system, and the method employs forward modelling as well as iterative and inverse techniques. I found myself having to refer back to earlier parts of the methods to remind myself how the information is all linked; it would be very useful for the reader if you could include a workflow diagram that documents all the steps in the methodology and indicates what type of information is used to quantify/constrain each component of the system.

**Data/Model Constraints**: there is very limited information on some of the data sets/model outputs used to constrain different components of the system and uncertainties are not explicitly quantified. For example, there is very little information on the firn densification model (FDM) used to constrain the 'surface' terms ($dg_{surf}$ in eq. 1 and $dh_{surf}$ in eq. 3) or the basal melt rate estimates used to determine the $dM_{melt}$ term in eq. 5. I was unable to identify how $dh_{if}$ (eq. 3) was constrained, this term describes ice mass loss/thinning due to dynamic ice flow processes.

**Spatial resolution**: the GRACE data have a spatial resolution of approximately 300 km, the altimetry data are resolved on a 100 x 100 km grid, and the spatial resolution of the glacial isostatic adjustment (GIA), FDM, and basal melt models are not stated. How did you deal with the different spatial resolution of these data sets/model outputs and to what degree do you think the spatial resolution of the data sets biases the final results? For example, do you think the spatial patterns in figures 3a and 4a accurately reflect the spatial variability of the basal water storage trends? Similarly, do you think the absolute magnitude of the inferred basal water storage trend is accurate or could the local signal be under-estimated due to the long wavelength nature of the data sets used?

**Iterative approach**: when calculating $dg_{BMB}$ (eq. 2), what initial values are assumed for the $dg_{ivm}$ term, and what is the magnitude of the correction that is applied when $dg_{BMB}$ is re-calculated after the initial solution is used to estimate the magnitude of $dg_{ivm}$? Related to this point: please provide additional information on the assumed relationship between basal mass balance and the vertical motion of the ice sheet (e.g. line 55).

**Temporal variability**: all calculations assume a linear rate of change over the duration of the study period (2003-2009). Please provide justification for this assumption. Could/should any of the processes be considered as time-varying, e.g. the surface mass components, and would you expect the basal mass balance to exhibit time-variability or a linear trend?

**Bed deformation**: how are the predicted basal uplift rates 'forced' (line 170)? I think you apply a method to ensure that they match GPS rates from 2003-2009; what are the uncertainties on the GPS rates, and can you confirm that they primarily reflect the GIA process, i.e. that they have been corrected for the elastic response to contemporary surface mass change? This raises a follow-on question: in your methodology, do you account for the elastic response to surface mass change, which will be recorded in both the gravity and (to a lesser degree) altimetry data?

**Uncertainties**: the results depend on the ability to accurately remove signals associated with glacial isostatic adjustment, surface mass balance processes, and basal melt. Uncertainties associated with these factors are briefly mentioned in section 4.1, along with mention of GRACE uncertainties (but not altimetry data uncertainties), but the magnitude of the uncertainty associated with each individual process is not quantified. This makes it difficult to assess how robustly each signal can be quantified and hence how accurately the basal water storage variation signal has been isolated. In particular, uncertainties in the firn densification model are invoked a number of times in section 4.2 to explain some surprising results shown in fig. 4, but the details are not quantified.

**Subglacial lakes**: you relate some of your results to subglacial lake behaviour (e.g. lines 13, 283). Given the difference between the spatial resolution of the satellite data (~100 km) and the typical size of a subglacial lake (~10 km), and the fact that lakes are documented to fluctuate in volume over just a few years (e.g. Fricker et al., 2016), it would be useful to include some discussion of how accurately you think your results reflect the behaviour of individual subglacial lakes.

**Amundsen Sea region**: I was unable to follow your argument on lines 204-206, which suggests that since the region inferred to have a negative trend in basal water storage (shown in fig. 3a) is similar to the region previously documented to be losing ice mass, this implies that mass loss in the region is "dominated by basal water discharging rather than surface mass loss". Could it be that some of the negative signal in fig. 3a reflects the incomplete removal of the signal due to dynamic ice thinning, a process that is not really mentioned in any detail? Similarly, on line 220, it is stated that "basal water decreases… are often accompanied by rapid ice flow". This seems a little odd because a reduction in basal water would increase friction at the bed and ultimately reduce ice flow; the paradox may be explained if the mass loss signal reflects ice thinning rather than a reduction in basal water storage.

**Non-aqueous mass variation**: please describe what processes are represented by the $dM_{non}$ term when it is first introduced in equation 5 (and maybe also describe the processes in the Introduction). When the processes are eventually described in section 4.2 more references are needed, e.g. to support the suggestion that negative values may reflect glacial erosion (line 261). Equation 5 seems to imply that basal non-aqueous mass variations only occur in regions where basal melt is zero. However, on line 244 it is stated that non-aqueous mass decreases may reflect ice discharge, a process that can take place where basal melt is non-zero. In general, if I have correctly interpreted eq. 5 to imply that $dM_{non}$ is only defined where basal melt is zero, I am confused as to how this quantity can be quantified for the whole of Antarctica (fig. 4a), given that there are large regions where basal melt is non-zero. Apologies if I have misunderstood the methodology here.

**Ice shelf effects**: it is not clear what process are described by the phrase "the blocking effect of the Ronne Ice Shelf" (lines 198, 227).